# Characterization of Kefir Produced in Household Conditions: Physicochemical and Nutritional Profile, and Storage Stability

**DOI:** 10.3390/foods10051057

**Published:** 2021-05-11

**Authors:** Emília Alves, Epole N. Ntungwe, João Gregório, Luis M. Rodrigues, Catarina Pereira-Leite, Cristina Caleja, Eliana Pereira, Lillian Barros, M. Victorina Aguilar-Vilas, Catarina Rosado, Patrícia Rijo

**Affiliations:** 1CBIOS—Research Center for Biosciences & Health Technologies, Universidade Lusófona’s, Campo Grande 376, 1749-024 Lisboa, Portugal; emilia.alves@ulusofona.pt (E.A.); p5999@ulusofona.pt (E.N.N.); joao.gregorio@ulusofona.pt (J.G.); monteiro.rodrigues@ulusofona.pt (L.M.R.); catarina.leite@ulusofona.pt (C.P.-L.); 2Department of Biomedical Sciences, Faculty of Pharmacy, University of Alcalá, Carretera Madrid-Barcelona, Km 33.100, 28805 Alcalá de Henares, Madrid, Spain; mvictorina.aguilar@uah.es; 3LAQV, REQUIMTE, Departamento de Ciências Químicas, Faculdade de Farmácia, Universidade do Porto, Rua de Jorge Viterbo Ferreira 228, 4050-313 Porto, Portugal; 4Centro de Investigação de Montanha (CIMO), Instituto Politécnico de Bragança, Campus de Santa Apolónia, 5300-253 Bragança, Portugal; ccaleja@ipb.pt (C.C.); eliana@ipb.pt (E.P.); lillian@ipb.pt (L.B.); 5Instituto de Investigação do Medicamento (iMed.ULisboa), Faculdade de Farmácia, Universidade de Lisboa, 1649-003 Lisboa, Portugal

**Keywords:** kefir, household conditions, storage time influence, nutritional composition, fatty acid profile, particle size, polydispersity index, zeta potential

## Abstract

Kefir, a traditional fermented food, has numerous health benefits due to its unique chemical composition, which is reflected in its excellent nutritional value. Physicochemical and microbial composition of kefir obtained from fermented milk are influenced by the type of the milk, grain to milk ratio, time and temperature of fermentation, and storage conditions. It is crucial that kefir characteristics are maintained during storage since continuous metabolic activities of residual kefir microbiota may occur. This study aimed to examine the nutritional profile of kefir produced in traditional in use conditions by fermentation of ultra-high temperature pasteurized (UHT) semi-skimmed cow milk using argentinean kefir grains and compare the stability and nutritional compliance of freshly made and refrigerated kefir. Results indicate that kefir produced under home use conditions maintains the expected characteristics with respect to the physicochemical parameters and composition, both after fermentation and after refrigerated storage. This work further contributes to the characterization of this food product that is so widely consumed around the world by focusing on kefir that was produced in a typical household setting.

## 1. Introduction

Traditional kefir has been consumed for centuries [1,2] due to its high nutritional value and is therefore considered a health promoting food [3]. Several health benefits have been attributed to kefir, mainly justified by the bioactivity of metabolites produced during fermentation [4,5], such as improved lactose digestion and tolerance [6], anti-inflammatory effect [7,8], antimicrobial activity [9], antioxidant activity [10], antitumor activity [11], wound healing [9,12], modulation of the immune system [13], and growth inhibition of pathogenic microorganisms [14,15]. Traditional kefir production uses kefir grains as starter culture, differentiating it from other fermented foods [16]. Kefir grains can maintain their activity as long as they are preserved and incubated under appropriate conditions, due to their extremely stable microbial composition [17,18,19]. The microorganisms usually found in the grains are homo and heterofermentative lactic acid bacteria (LAB), Lactobacillaceae family (genera *Lactobacillus* and *Leuconostoc)* and Streptococcaceae family (genera *Lactococcus* and *Streptococcus*), acetic acid bacteria Acetobacteraceae family (genera *Acetobacter*) and yeasts Saccharomycetaceae family (genera *Kluyveromyces* and *Saccaromyces*) [20,21,22,23,24]. The viability of kefir grains is guaranteed through maintenance of the bacterial/yeast ratio achieved by continuous fermentation cycles that lead to their biomass increase [2,20,25]. This increment is dependent on temperature, pH, washing of grains, renewal of milk, and presence of nutrients [16,25,26,27]. Grain preservation for household kefir production can be achieved either by continuous fermentation cycles, and assured for ten weeks of propagation [17], or by freezing at −20 °C [19]. The microbiological composition of kefir grains depends on their origin [20,28,29,30].

Kefir’s microbiota is different from that of the grains [16]. The physicochemical and microbial composition of kefir fermented milk is influenced by the type of the milk, grain to milk ratio, time and temperature of fermentation, and storage conditions [31,32,33,34,35]. Traditional kefir typically uses cow’s milk as substrate [30,33]. Although whole, semi-skimmed, or skimmed milk can be used [32,36], the latter creates a kefir with significantly lower nutritional quality [29]. Grain to milk ratio, usually varying between 2% and 10% (*w*/*v*), influences the kefir microbial profile, and higher rates of grain inoculum increase lactic acid levels, providing sharper pH lowering [14,31,36]. The viscosity is also affected, since higher percentages of kefir grain inoculate produce a more acidic, but less viscous, kefir [31,36]. Lactose content is the main nutritional compound influenced by the amount of grain inoculum and smaller ratios of inoculate results in kefir with higher lactose levels [21,36].

Typical kefir fermentation occurs at temperatures between 20 and 25 °C for approximately 24 h, with pH varying between 4.2 and 4.6 [14,31,37]. During fermentation, the chemical composition of kefir changes mainly due to lactose conversion by homofermentative LAB, first into lactic acid, causing the pH to drop and acidity to increase [21,38], followed by the remaining hydrolyzation into glucose and galactose by the enzymatic activity of β-galactosidase present in the grains [39]. Further into the fermentation cycle heterofermentative LAB convert glucose into CO_2,_ ethanol and lactic acid, the latter being the most predominant organic acid after fermentation, and in this environment proteins are converted into peptides [34,36,40,41]. The production of lactic acid contributes to the antimicrobial effect of kefir, and since it acts as a natural preservative, allows the homemade product to have a low contamination risk [14,24,42].

The chemical composition of kefir reflects its nutritional value and the recommended quality standards for kefir are at least 2.8% protein, less than 10% fat, and at least 0.6% lactic acid [43]. Kefir can be consumed immediately after grain separation or may be refrigerated for later consumption [4,25,41]. Fermented milk characteristics must be maintained during storage; however, since continuous metabolic activities of residual kefir microbiota may occur, the composition of refrigerated kefir may be affected during storage [19,34,36,44]. Kefir can maintain a shelf life of 3–12 days [25]. During refrigerated storage at 4 °C, viscosity is reported to decrease abruptly with time [34,36], while total fat, lactose, dry matter, and pH remain constant until 14 days of storage [32,36,40] and lactic acid slightly increases after 7 days storage [40]. Although the lipolytic activity in milk fat by LAB is limited, it can still contribute to the production of free fatty acids [45].

This work aimed to study kefir produced in representative household conditions, characterizing the properties, nutritional composition, and stability of a freshly made and 48 h refrigerated beverage. To date, information on homogeneity and stability of traditional kefir is scarce, and to the best of our knowledge, this is the first study to provide information on kefir produced by simulating representative home use conditions. The innovative character of this study also resides in the fact that, in addition to the usual parameters for physicochemical analysis and composition, the fatty acid profile, particle size, polydispersity index (PdI), zeta potential, and Fourier Transform Infrared Spectroscopy (FTIR) spectra analysis were also included in the global evaluation of kefir.

## 2. Materials and Methods

### 2.1. Kefir Grains Storage and Kefir Production

Kefir grains CIDCA AGK1 were obtained from the Centro de Investigacíon y Desarrollo en Criotecnologia de Alimentos (CIDCA), La Plata, Argentina. Microbiological characterization of these grains has been described elsewhere [20,46,47]. Kefir grains were maintained in milk and preserved by storage in a freezer at −20 ± 2 °C, which proved to be the best method for grain preservation and can also be used to maintain the grains for household kefir production [19]. After this type of storage, the grains were activated before use in fermentation [27]. For activation, grains were left to defrost at room temperature for 12 h, after which they were placed in semi-skimmed milk at 20 ± 1 °C for 24 h. The activation step was repeated three times.

Kefir beverage samples were produced by fermentation for 24 h of a commercial ultra-high temperature pasteurized (UHT) semi-skimmed cow milk of Portuguese provenance (Nova Açores^®^, S. Miguel, Portugal), with CIDCA AGK1 kefir grains using a grain inoculum of 10% (*w*/*v*), at a temperature of 20 ± 1 °C. After fermentation, grains were separated from the fermented milk by filtration through a plastic sieve and used as starter culture for the next kefir batch, under the same conditions. Samples of fermented milk kefir were collected after filtration.

### 2.2. Activity of Kefir Grains

#### 2.2.1. Biomass Growth

Kefir grain biomass increase was measured over 8 days, with daily inoculations in milk. Kefir grains were sub-cultured by successive passage of the total amount of grains in increasing volumes of milk to maintain a concentration of 10% (*w*/*v*) [19]. Rising of the grains was made with milk, because growth of the grains is retarded when they are rinsed with water after each sieving [27]. Kefir grains were separated from fermented milk by filtration using a plastic sieve. Grains were rinsed with milk at room temperature and left to dry on a filter paper at room temperature (20 ± 1 °C), after which kefir grains were weighed using an analytical scale (KERN ALJ220-4NM (KERN & Sohn GmbH, Balingen, Germany)) for gravimetric determination. After weighing, kefir grains were used as a new inoculum, maintaining the grain to milk ratio. Samples were made in duplicate. Biomass growth rate was determined gravimetrically, and increment percentage was calculated. All measurements were made in triplicate.

#### 2.2.2. Acidification Kinetic

Kefir fermentation of milk was carried out at 20 ± 1 °C and samples of fermented milk were collected every 2 h until a stabilized pH value was reached. Measurements were made using a Metrohom 827 pH lab^®^ digital meter (Metrohom AG, Herisau, Switzerland). Samples were made in duplicate. All measurements were made in triplicate.

### 2.3. Viable Microorganisms and Inhibitory Activity Test

#### 2.3.1. Bacterial and Yeasts Counts

Determination of LAB and yeast counts were made by conventional culture techniques [48]. Tryptone water (Sigma-Aldrich, St. Louis, MO, USA) at a concentration of 1 g/L was used to prepare the dilutions for the microbiological analyses. Ten-fold dilutions in 0.1% sterile tryptone water were plated in each medium. LAB counts were quantified on De Man, Rogosa and Sharpe (MRS) agar plates (Oxoid, Hampshire, UK), which were incubated at 30 ± 1 °C for 24 h, and then for another 24 h, under the same conditions. Yeast counts were quantified on Yeast-Extract Glucose Chloramphenicol (YGC) agar plates (Oxoid, Hampshire, UK), which were incubated at 30 ± 1 °C for 48 h. Counts were expressed in total colony-forming units per milliliter. Measures were made in duplicate.

#### 2.3.2. Inhibitory Activity Test

Inhibitory activity of kefir fermented milk was evaluated on the growth of *Escherichia coli* using conventional culture techniques [48]. *E. coli* counts were quantified on Eosin Methylene Blue Agar (EMB) agar plates (Oxoid, Hampshire, UK), which were incubated at 37 ± 1 °C for 24 h. A known *E. coli* strain (ATCC 25922) was used as control. Measures were made in duplicate.

### 2.4. Physicochemical Characteristics of Kefir Beverage

#### 2.4.1. Particle Size, Polydispersity Index, and Zeta Potential

The particle size and PdI were analyzed, by dynamic light scattering, for milk (control), kefir immediately after the 24 h fermentation period (t0), and kefir after storage at 5 ± 1 °C for 24 and 48 h (t24 and t48, respectively). Zeta potential was also evaluated at the same time points by an electrophoretic mobility technique using a Delsa™ Nano C from Beckman Coulter, Inc. (Brea, CA, USA). All analyses were run in triplicate at room temperature (20 ± 1 °C) after diluting the samples with distilled water. Dilutions of 1:50 or 1:125 were used in the case of unfermented or kefir fermented milk, respectively.

#### 2.4.2. Fourier Transform Infrared Spectroscopy (FTIR)

Kefir samples (t0, t24, and t48) obtained after freeze-drying were evaluated by FTIR in a PerkinElmer^®^ Spectrum 400 (PerkinElmer Inc, Waltham, MA, USA) equipped with an attenuated total reflectance (ATR) device. The ATR system was cleaned before each analysis by using dry paper and scrubbing it with methanol and water (50:50). The room air FTIR-ATR spectrum was used as background to verify the cleanliness and to evaluate the instrumental conditions and room interferences due to H_2_O and CO_2_. The spectra were obtained collecting 100 scans of each sample, between 4000 and 600 cm^−1^, with a resolution of 4 cm^−1^. The FTIR analysis was also performed for unfermented milk as a control sample.

#### 2.4.3. Viscosity and pH

The viscosity and pH were evaluated for both control and kefir samples (t0, t24, and t48). Viscosity was measured using a Brookfield Ametek DV3T^®^ rheometer (AMETEK Brookfield, Middleboro, MA, USA) with a SV18 spindle, at 30 rpm. Measurements were performed at 25.2 ± 0.2 °C and readings were recorded for 1 min. pH was measured using a Metrohom 827 pH lab^®^ digital meter (Metrohom AG, Herisau, Switzerland). Samples were made in duplicate, and all measurements were made in triplicate.

### 2.5. Nutritional Analysis of Kefir Beverage

For chemical analysis, both control and kefir samples (t0, t24, and t48) were frozen at –80 °C for 24 h, after which all samples were freeze-dried in a Labconco FreeZone 25^®^ (Labconco, Kansas City, MO, USA) using a surface condenser temperature of −50 °C and 400 mTorr for 24 h. Samples were weighed before freezing and after freeze-drying for mass determination. The contents of protein, fat, carbohydrates, and ash were determined according to the official analysis methodologies AOAC [49] and following a procedure previously reported by Barros et al. [50]. Protein was determined considering the total nitrogen content and using the specific conversion factor for milk (6.38). Total fat content was analyzed as fatty acids and expressed as triglyceride equivalents. Ash was determined by gravimetry. Total carbohydrate content was determined by difference, as follows: 100-(weight in grams (protein + fat + water + ash + alcohol) in 100 g of food) [51]. Dry matter was calculated as the sum of total fat, protein, ash, and carbohydrates content. Total energy was calculated following the Equation:Energy (kcal) = 4 × (g protein + g carbohydrates) + 9 × (g fat).

All samples were also evaluated regarding the sugar content, following an extraction procedure previously described [50]. Samples were then filtered through 0.2 μm Whatman nylon filters into a 1.5 mL vial for liquid chromatography analysis. The HPLC system was coupled to a refraction index (RI) detector and the free sugars were identified by comparison with standards and further quantified considering the internal standard and results were expressed in g per 100 g [50]. In addition, all samples were also evaluated for fatty acids content, which were extracted from all the samples and determined by gas chromatography coupled with a flame ionization detector (GC-FID, DANI model GC 1000, Contone, Switzerland) using a procedure previously described by Barros et al. [50]. The results were expressed as relative percentage of each fatty acid (%). Two batches of all samples were made, and all analyses were performed in duplicate.

### 2.6. Statistical Analysis

Results were expressed as mean ± standard deviation (SD). Linear regression was used to assess grains biomass growth. Differences over the groups were identified using one-way ANOVA analysis of variance. Different letters show significant differences by Tukey Post hoc multiple comparison tests. When homogeneity was not guaranteed, Games-Howell post-hoc tests were used. All analyses were performed using the SPSS statistical package version 25 (SPSS Inc., Chicago, IL, USA) with a level of significance of 0.05.

## 3. Results

### 3.1. Activity of Kefir Grains

#### 3.1.1. Biomass Growth

Biomass growth of kefir grains, incubated at 20 ± 1 °C, for successive 24 h periods over 8 days, expressed in weight (g), is showed in Figure 1.

Our results showed that after 8 days of successive fermentations the biomass grains had an increment of 60% when compared to the initial weight of the grains. We found that our CIDCA AGK1 grains had a mean 24 h biomass growth of 6 ± 2%, after fermentation at 20 °C.

#### 3.1.2. Acidification Kinetic

The acidification rate of milk measured during kefir fermentation are showed in Figure 2.

As expected, during fermentation the pH value of kefir dropped from the value of 6.6 of unfermented milk, reaching a mean value of 4.5 ± 0.1 at the end of 24 h. After 42 h, the mean pH value of the kefir beverage stabilized at 3.9 ± 0.1.

### 3.2. Viable Microorganisms and Inhibitory Activity Test

The viable LAB and yeast counts, as well as coliforms found in the kefir analysis are presented in Table 1.

The microbiological analysis of our kefir revealed 7 × 10^7^ CFU/mL of LAB and 2 × 10^6^ CFU/mL of yeast. Furthermore, the absence of coliforms (*E. coli*) was also confirmed (Table 1).

### 3.3. Physicochemical Characteristics of Kefir Beverage

#### 3.3.1. Particle Size, PdI, and Zeta Potential

The hydrodynamic diameter, PdI, and zeta potential of unfermented milk and kefir beverages, according to storage conditions, are presented in Table 2. In all cases, nanometric diameters (250–439 nm) were found for all beverages, with PdI values lower than 0.3, and zeta potential values smaller than −30 mV. Kefir at t0 showed a particle size and a PdI significantly higher (*p* < 0.0001 and *p* < 0.0001, respectively) than control, but no statistical difference was observed for zeta potential (*p* = 0.483). 24 h refrigerated kefir presented a smaller particle size (*p* < 0.0001), a smaller PdI (*p* = 0.001) and also a smaller zeta potential (*p* = 0.013) compared to kefir at t0, but no differences were observed throughout storage for these parameters (*p* = 0.975, *p* = 0.575, and *p* = 0.996, respectively).

#### 3.3.2. FTIR

FTIR spectra were collected for unfermented milk and for kefir samples (t0, t24, and t48) (Figure 3). The control spectrum showed the presence of a broad band at 3335.99 cm^−1^: it was attributed to -ΟΗ stretching in hydroxyl groups associated with carbohydrate structures. The peaks at 2915.49 cm^−1^ are associated with C-H bending in fatty acids, 1639.26 cm^−1^ correlates to the carbonyl (C=O) stretching or N-H and C-H bending vibration of the milk proteins. The band2 2915 and 2856.7 cm^−1^ may be due to the anti-symmetric and symmetric stretching of CH_2_ groups from the fatty milk components.

From Figure 3 we can observe a very strong overlap between the spectral signals of milk (control) and kefir samples: t0, t24, and t48. This is evident throughout the full-recorded spectral region, suggesting a high similarity in the composition between the samples.

#### 3.3.3. Viscosity and pH

After a 24 h fermentation, kefir showed a mean pH value of 4.60 ± 0.05, which was significantly lower than that of the control (*p* < 0.0001). No statistical difference in pH values was observed between samples t0 and t24 (*p* = 0.116) and between both refrigerated samples (*p* = 0.168). However, the pH value decreased between t0 and t48 (*p* = 0.014). Kefir at t0 showed a mean viscosity of 32 ± 4 mPa.s, which was significantly higher than that of the control (*p <* 0.0001). The viscosity of kefir decreased after a refrigerated storage period of 24 h (*p* = 0.043). However, no significant difference in viscosity was observed between both refrigerated samples (*p* = 0.732) (Table 3).

### 3.4. Nutritional Analysis of Kefir Beverage

The nutritional content, evaluated by fat, protein, carbohydrates, ash, lactose, and lactic acid content, as well as the energy value of unfermented milk and the kefir samples immediately after 24 h fermentation at 20 °C, and after 24 h and 48 h of cold storage at 5 ± 1 °C, is shown in Table 4.

Kefir at t0 showed a mean nutritional composition of 1.28 ± 0.04 g/100 mL of fat, 3.15 ± 0.19 g/100 mL of protein and 4.91 ± 0.19 g/100 mL of carbohydrates. As macronutrients are concerned no difference was observed in fat, protein, and carbohydrates content due to fermentation or storage (*p* = 0.071, *p* = 0.071 and *p* = 0.449, respectively). Energy, ash, and dry matter (DM) were different between t0 and control (*p* = 0.002, *p* = 0.011 and *p* = 0.028, respectively). Finally, the lactose content in control was significantly higher than in kefir (*p* = 0.011) with a 13.6% decrease during fermentation. Consequently, the lactic acid content in kefir was significantly higher than that of the control (*p* = 0.001). No differences were found between kefir samples for lactose and lactic acid (*p* = 0.100 and *p* = 0.580, respectively).

The content of fatty acids of unfermented milk and kefir beverages was also determined, and the results are presented in Table 5. All samples evidenced the presence of 18 fatty acids, comprising saturated (SFA), monounsaturated (MUFA), and polyunsaturated (PUFA) fatty acids.

SFA content of kefir at t0 was significantly lower than control (*p* = 0.012), which is supported by the differences observed between the samples regarding palmitic acid content (*p* = 0.029). No differences were observed for the remaining SFA, between these samples. Within SFA, palmitic acid (C16:0) stands out as the major fatty acid followed by and myristic acid (C14:0). MUFA content of kefir at t0 was significantly higher than that of control (*p* = 0.008), which may be mainly supported by the difference observed between the samples regarding oleic acid content (*p* = 0.014). No differences were observed for the remaining MUFA, between these samples. Within MUFA, oleic acid (C18:1n-9) represents the major component. PUFA content showed no difference between all samples (*p* = 0.050), which may be supported by the fact that linoleic acid (C18:2n-6), the major PUFA component, also remained constant (*p* = 0.083), despite the content of α-linolenic acid (C18: 3n-3) increased slightly after fermentation (*p* = 0.023).

Between t24 and t0, no difference was observed in SFA and MUFA content (*p* = 0.389 and *p* = 0.460, respectively). Within SFA, only C8:0 evidenced a very small decrease (*p* = 0.041), while within MUFA no change was observed. Between t48 and t0, no difference was observed in the SFA content (*p* = 0.083). Within SFA, lauric acid (C12:0), pentadecylic acid (C15:0), palmitic acid and margaric acid (C17:0) presented a very small increase (*p* = 0.022, *p* = 0.009, *p* = 0.002 and *p* = 0.048, respectively). MUFA content decreased (*p* = 0.031), with only palmitoleic acid (C16:1) reflecting that decrease (*p* = 0.002).

## 4. Discussion

Kefir grains are traditionally cultured in milk at room temperature, which is considered to be between 20 and 25 °C [36,41]. The traditional use, combined with the fact that 20 °C is in the range of typical indoor conditions of a Portuguese house [52], thus reflecting the domestic scenario of preparation of kefir, justifies the choice of the fermentation temperature in our study. It is widely known that biomass increase and lactose consumption rise at higher incubation temperatures [37]. Nevertheless, Londero et al. [53] found that biomass growth, acidification capacity, and maintenance of the chemical composition are optimized at a fermentation temperature of 20 °C.

Increment of the grains biomass during fermentation highlights the microbial growth resulting of the balance within the microbiota of the grains [25,53,54]. This biomass increase is mainly due to the production of protein and polysaccharides by its microbiota within the grains matrix, which can be transferred to the fermented milk [54]. Our grains presented a mean biomass increment of 6 ± 2% after a 24 h fermentation of semi-skimmed milk at 20 °C, which is consistent with the results of DeSainz et al. [55], that found a biomass increase of 7.2 ± 0.1% after 24 h fermentation at 35 °C. Interestingly, using a mathematical model Zajšek and Goršek [37] observed a linear trend between fermentation temperature and increase of biomass grains, that predicted an increase of 7.034 g/L in biomass grain growth for a temperature of 20 °C. Our results showed a ten-fold higher growth, which shows a considerable disparity between a mathematical model and a real fermentation scenario. The growth behavior of our grains (Figure 1) was contrary to the results found by Pop et al. [56] using a grain inoculum of 4.5% (*w*/*v*) to ferment skimmed milk at 25 °C and showing a significant biomass decrease after 24 h. This may be justified by the fact that our study used semi-skimmed milk, thus making Pop’s justifications, nutrient depletion or increase acidity, less robust arguments to justify growth behavior of our grains. Moreover, the fat content of the milk may be of significant importance, as demonstrated by Schoevers and Britz [27], who reported that higher milk fat content impairs grain growth by inhibition of nutrient exchange. The authors also found that the lowest increase in biomass happened when low fat milk was used, and their results using this milk type and a grain to milk ratio of 1% (*w*/*v*) showed a biomass increase around 50% after 8 days [27]. The increase of 60.07% in biomass that we found may also be justified by the use of a grain inoculum of 10% (*w*/*v*).

The mean pH value of 4.5 ± 0.1 that was verified after 24 h of fermentation is in agreement with that found by Garrote et al. [31], using the same type of grain inoculum. The acidification rate observed during fermentation in our work (Figure 2) is consistent with the literature [21,31,36,37,41] and may reflect the LAB capability to acidify the milk [37,41]. Both pH and lactic acid variation during fermentation of kefir represent an indirect measure of the biological activity of the grains [57]. LAB population present high sensitivity to low pH values, which contributes to their decline, being that the main reason why kefir does not become more acidic through time [31,36].

Interestingly, despite the home use production conditions, the resulting kefir (Table 1) is in conformity with the recommendations of *Codex Alimentarius* for fermented milks (Codex Stan 243-2003), thus complying with a number of total micro-organisms of at least 10^7^ colony-forming units (CFU)/mL and a yeast number not less than 10^4^ CFU/mL [43].

After fermentation, we found that the mean particle size of kefir (439 ± 42 nm) predictably increased significantly compared to the unfermented milk (280 ± 54 nm) and decreased again after 24 h-refrigerated storage (256 ± 6 nm), remaining stable for another 24 h of cold storage (249 ± 1 nm). According to the literature [58], casein micelles aggregation is promoted by increase of acidification, protein content, fat content and temperature, thus these factors may directly affect particle growth in kefir beverage. The pH decrease observed in freshly made kefir (Table 3) may be at the root of the initial aggregation of casein micelles into larger clusters. After refrigerating the kefir beverage for 24 h, the size of casein micelles probably decreased due to the effect of low temperatures on protein aggregation. In fact, it was already reported that the higher the temperature, the higher the particle size of fermented milk [58]. Moreover, it is noteworthy that, after 24 h of refrigerated storage, the particle size of kefir is in reasonable agreement with the results recently presented by Beirami-Serizkani et al. [59]. Another 24 h of refrigerated storage did not alter the particle size of kefir, probably due to the fact that, during this period, the temperature remained constant, as well as no pronounced alterations were found in pH values (Table 3) and protein or fat content (Table 4) of the kefir beverage.

The degree of non-uniformity of a population’s size distribution within a given sample, represented by PdI, suggests the degree of heterogeneity of the sample. A homogeneous sample, perfectly uniform regarding the particle size, shows a PdI value of zero, while a heterogeneous sample, highly polydisperse with multiple particle size populations has a PdI of 1 [60]. The stability of a sample, given by the zeta potential, is a measure of the magnitude of electrostatic repulsion/attraction or charges between particles [58] and increases with the homogeneity of the size distribution [60]. Zeta potential depends on factors like temperature, acidity, and viscosity, and a highly negative/positive zeta potential foresees a more stable dispersion, while values lower than |30| mV can indicate colloidal instability, which can lead to aggregation [61]. Concerning the particle size distribution of the analyzed samples, given by PdI (Table 2), it is remarkable that all beverages display uniform particle size distributions (PdI < 0.3). Despite that, the increase in particle size of kefir in comparison with unfermented milk also resulted in an increase of PdI, which was almost recovered by the decrease of particle size upon refrigerated storage for 24 h and 48 h (Table 2). In addition, the zeta potential values recorded for all samples (<−30 mV, Table 2) indicate that all beverages display good colloidal stability. It is noteworthy that the zeta potential of unfermented milk was in line with a previous report of its variation with milk pH [62]. According to our results, the zeta potential of kefir is similar to that of unfermented milk, slightly increasing with refrigerated storage (Table 2). This is not in agreement with the data reported by Beirami-Serizkani et al. [59], showing that the different preparation procedures of kefir drinks may influence the colloidal stability of the resulting beverage.

FTIR spectrum analysis of unfermented semi-skimmed milk (Figure 3) was consistent with the literature [63]. Using FTIR spectra, we confirm that the physicochemical properties of the milk change during the fermentation process. However, from the strong overlap between the kefir spectral signals (Figure 3) we corroborate that its physicochemical properties are maintained during refrigerated storage, which is consistent with the results obtained from the other analysis performed in this study.

The variations in pH and viscosity found in our kefir samples (Table 3) are similar to those reported the literature [21,36,44]. The pH value of kefir was significantly lower than that of milk, remaining constant in the first 24 h of refrigeration and showing a slight decrease of 2% at the end of 48 h (Table 3). Similar results after 2 days of storage were reported by Leite et al. [21]. Irigoyen et al. [36] also reported no variations in pH during kefir storage, and attributed it to the presence of yeast in the grains, since the production of lactic acid by LAB is slower in the presence of yeasts than in pure culture [38,44].

After a 24 h fermentation, kefir revealed a significantly higher viscosity compared to the unfermented milk (Table 3). This can be in part attributed to the production of kefir’s exclusive polysaccharide, kefiran, which, in addition to constituting the grain structure, can also be found dissolved in the liquid, thus contributing to the rheology of the fermented beverage [64]. The decrease observed in kefir’s viscosity after the first 24 h refrigerated storage period (Table 3) can be attributed to the hydrolysis of the polysaccharide kefiran together with the reduction observed in the LAB responsible for the polysaccharide’s production [34]. Throughout storage, a decrease in viscosity and phase separation (syneresis), due to the aggregation of casein micelles and subsequent precipitation are the most typical events that may impair the quality of kefir [65]; however, these changes only become evident in periods of storage longer than seven days [34,36,44]. Nevertheless, our data showed no difference in viscosity during storage, which may be attributed to a limited storage time (only 48 h).

The nutritional composition of kefir is influenced by milk composition, origin of the grains, temperature, and duration of fermentation and storage conditions [31,36]. As explained previously, our kefir prepared in a typical home use setting fulfills the requirements the *Codex Alimentarius* (Table 4) and is in accordance with data reported by other authors [21,36,66]. Whilst typical cow milk presents a carbohydrate content between 4.7 and 4.9 g/100 mL, reflecting essentially lactose content [67], kefir has a carbohydrate content around 11.9 g/100 g, also reflecting the presence of polysaccharide kefiran [54]. Our data for unfermented milk were consistent with the literature [67], but no difference was observed in carbohydrate content, neither during fermentation or storage. It is noteworthy that in spite the small lactose decrease observed after fermentation, the carbohydrate profile of kefir is expected to be different from that of the source milk, due to the presence of polysaccharide kefiran in kefir (not quantified in this study).

After 24 h fermentation we observed a decrease of 13.6% in lactose level and an increase in lactic acid content which is consistent with the literature [21,33,36,38,44], and may be explained by the hydrolysis of lactose and production of lactic acid in the initial LAB lactose metabolism [21,44]. These results are in line with those reported by Irigoyen et al. [36], who observed a 20–25% decrease in lactose during 24 h fermentation. Assadi et al. [68] reported much lower levels of lactose after 24 h of fermentation even though producing identical content of lactic acid. Throughout the storage period no changes were observed in lactose and lactic acid content of kefir, which is consistent with results reported for similar time storage [36,40,44]. Guzel-Seydim et al. [40] reported that during cold storage of kefir, lactic acid production may be impaired possible due to the decrease of LAB concentration attributed to pH drop [31,36]. Diversity in results involving lactose degradation and lactic acid production, in kefir fermentation, may be attributed to differences in grain to milk ratio and in different origins of kefir grains [21].

Even though, our data did not reveal any changes in fat, protein; and carbohydrates content, a small decrease in energy content was observed between milk and kefir, possibly due to variation of carbohydrates and fat, despite no statistical significance was found.

DM in freshly made kefir may range between 9.4% and 11.1%, and it is expected to change accordingly with the variation of fat and lactose comparatively with the source milk [36,38]. Our data are consistent with these, once we observed a slightly decrease of DM content after fermentation, which is consistent with the lactose variation also observed. Assadi et al. [68] observed only 5.56% of DM in kefir, however their value was also consistent with the much lower lactose level they found compared with the source milk. After 48 h storage, no differences were observed for both lactose and total fat and consequently also for DM. However, Irigoyen et al. [36] reported a DM content decrease after 48 h storage which is consistent with the fat content decrease verified in their study.

Milk proteins are affected by proteolytic activity of the kefir grains, producing different peptides and nonprotein nitrogen compounds, thus contributing to the protein profile of kefir [69]. However, during fermentation and storage, casein content does not change significantly, suggesting a low degree of casein proteolysis, contrary to the nonprotein nitrogen compounds derived from whey protein, that increase both in fermentation and in storage [70]. Even though the protein profile has not been determined in our work, its results are hereby supported, since no differences in the total protein content of kefir and unfermented milk were observed (Table 4). Moreover, utilization of protein nitrogen by bacteria during fermentation is limited, since their preferential energy source are carbohydrates [71]. Contrary results were reported by Vieira et al. [32], showing an increased protein level during fermentation, which were explained by the interaction between stress response proteins and lipid membrane unsaturation in bacterial cells, since fermentation is a stress factor for LAB [32].

Total fat composition of kefir was identical to that of the source milk (Table 4), which is consistent with the literature [32,36], and also no difference was observed during refrigerated storage [32,36,40]. However, the fatty acid profile of freshly made and refrigerated kefir differs (Table 5). Kefir at t0 presented a decrease of 2% in SFA and an increase of 5% MUFA, these variations being identically reflected in the content of palmitic acid (C16:0) and oleic acid (C18:1n-9), respectively. These results are in line with the literature [32,72] and are useful in order to consolidate the potential health benefits of kefir [73]. Vieira et al. [32], justified the change in SFA and PUFA with the increase of desaturase activity of LAB during fermentation [74] since the conversion ratio of saturated into unsaturated fatty acids can be attributed to desaturase activity [75]. Even though, in our data, PUFA content showed an increase after fermentation, the difference was not statistically significant. PUFAs are known to affect the aroma profile of kefir, and since an increase of PUFA would lead to a loss of the typical scent [76], it is confirmed that in our particular setting conditions the olfactive characteristics of kefir are maintained. In the first 24 h of refrigerated storage no change in fatty acids profile was noted, and after 48 h storage, only a slightly decrease in MUFA was observed. Contrary results were found by Vieira et al. [32], who reported higher MUFA and lower SFA content during storage, which was attributed to the ability of LAB to increase the production of free fatty acids by lipolysis of milk fat during the cold storage [77]. The differences observed in kefir´s fatty acids profiles, according to other authors, may be justified by the different origin of the grains since each bacterial community may present a unique fatty acids production [21,32].

## 5. Conclusions

Our results showed that the kefir produced under home use conditions using UHT milk is able to fulfill the *Codex Alimentarius* requirements and maintains its characteristics with respect to the physicochemical composition, both after fermentation, as well as during 48 h of refrigerated storage. Whereas fat, protein; and carbohydrate content suffered no significant changes over fermentation, lactic acid increased, and lactose decreased, as expected. The fatty acids profile of the milk and kefir samples changed during fermentation revealing a decrease in SFA, an increase in MUFA, and no change in PUFA. Refrigerated storage did not significantly impact nutritional composition and fatty acids profile, thus attesting for the stability of kefir under these conditions.

To the best of our knowledge, this is the first study to aggregate information on detailed composition, homogeneity; and stability after refrigeration, of kefir produced using CIDCA AGK1 grains in a traditional in use setting. This work further contributes to the characterization of this food that is so widely consumed around the world by focusing on kefir that was produced in typical home use conditions.

## Figures and Tables

**Figure 1 foods-10-01057-f001:**
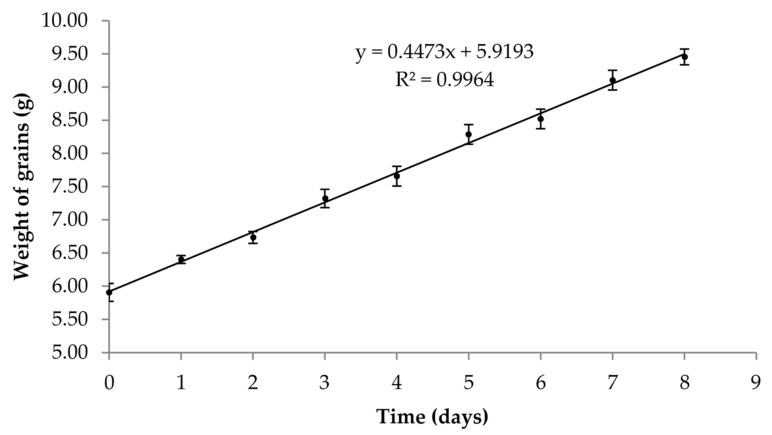
Increment of kefir grains biomass (g), incubated at 20 °C for 24 h periods, over 8 days (mean ± SD, *n* = 3).

**Figure 2 foods-10-01057-f002:**
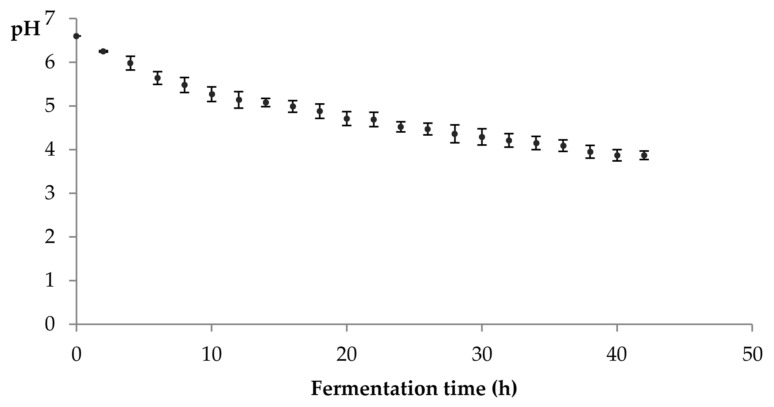
Acidification rate during fermentation at 20 °C, until pH stabilization (mean ± SD, *n* = 3).

**Figure 3 foods-10-01057-f003:**
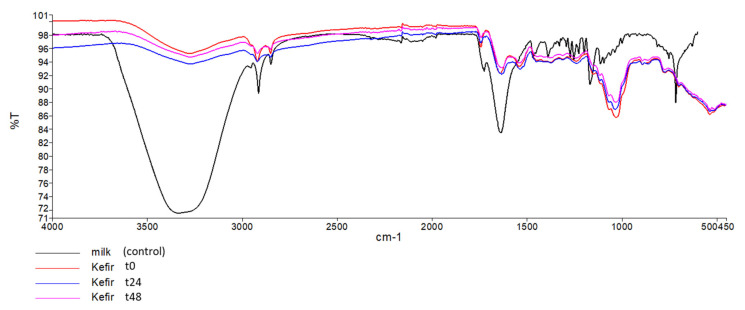
FTIR spectra of control and kefir samples (t0, t24, and t48).

**Table 1 foods-10-01057-t001:** Viable LAB and yeast counts (CFU/mL) and coliforms (CFU/mL) of kefir made from CIDCA AGK1 grains.

	Kefir Beverage
LAB (CFU/mL)	7 × 10^7^
Yeasts (CFU/mL)	2 × 10^6^
Coliforms (CFU/mL)	Absent

**Table 2 foods-10-01057-t002:** Hydrodynamic diameter, PdI, and zeta potential of control and kefir samples (t0, t24, and t48) (mean ± SD, *n* = 3).

	Control	Kefir		
		t0	t24	t48
Diameter (nm)	280 ± 5 ^b^	439 ± 42 ^a^	256 ± 6 ^b^	249 ± 1 ^b^
PdI	0.18 ± 0.01 ^b^	0.295 ± 0.006 ^a^	0.231 ± 0.008 ^c^	0.22 ± 0.02 ^c^
Zeta potential (mV)	−35 ± 2 ^a^	−38 ± 1 ^a^	−31 ± 2 ^b^	−30 ± 3 ^b^

^a–c^ Means within the same row with different superscripts are significantly different *p* < 0.05.

**Table 3 foods-10-01057-t003:** Physical parameters of control and kefir samples (t0, t24, and t48) (mean ± SD, *n* = 6).

	Control	Kefir		
		t0	t24	t48
pH	6.60 ± 0.00 ^a^	4.60 ± 0.05 ^b^	4.54 ± 0.02 ^b,c^	4.50 ± 0.04 ^c^
Viscosity (mPa.s)	2.11 ± 0.01 ^b^	32 ± 4 ^a^	26 ± 2 ^c^	24 ± 4 ^c^

^a–c^ Means within the same row with the different superscripts are significantly different *p* < 0.05.

**Table 4 foods-10-01057-t004:** Nutritional composition of control and kefir samples (t0, t24, and t48) (mean ± SD, *n* = 4).

	Control	Kefir		
		t0	t24	t48
Energy (kcal/100 mL)	48.2 ± 0.4 ^a^	43.8 ± 0.6 ^b^	44.5 ± 0.8 ^b^	44 ± 2 ^b^
Carbohydrates (% *w*/*v*)	5.14 ± 0.08	4.9 ± 0.2	5.0 ± 0.1	5.0 ± 0.2
Lactose (% *w*/*v*)	4.74 ± 0.05 ^a^	4.1 ± 0.2 ^b^	3.75 ± 0.08 ^b^	3.8 ± 0.2 ^b^
Proteins (% *w*/*v*)	2.8 ± 0.1	3.2 ± 0.2	3.1 ± 0.1	3.15 ± 0.05
Total Fat (% *w*/*v*)	1.81 ± 0.03	1.28 ± 0.04	1.32 ± 0.09	1.3 ± 0.3
Lactic acid (% *w*/*v*)	0.02 ± 0.00 ^b^	0.59 ± 0.07 ^a^	0.63 ± 0.01 ^a^	0.61 ± 0.05 ^a^
Ash (% *w*/*v*)	0.50 ± 0.01 ^a^	0.58 ± 0.02 ^b^	0.59 ± 0.01 ^a,b^	0.59 ± 0.02 ^b^
Dry matter (% *w*/*w*)	10.28 ± 0.04 ^a^	9.9 ± 0.1 ^b^	10.05 ± 0.09 ^b^	10.0 ± 0.2 ^b^

^a,b^ Means within the same row with different superscript letters show significant statistical differences (*p* < 0.05).

**Table 5 foods-10-01057-t005:** Fatty acids profile of control and kefir samples (t0, t24, and t48) (relative frequency, mean ± SD, *n* = 4).

	Control	Kefir		
Fatty Acids (%)		t0	t24	t48
C6:0	3.86 ± 0.05	3.6 ± 0.2	3.3 ± 0.1	3.7 ± 0.1
C8:0	2.12 ± 0.03 ^a^	2.07 ± 0.07 ^a^	1.95 ± 0.05 ^b^	2.18 ± 0.05 ^a^
C10:0	4.41 ± 0.02	4.5 ± 0.3	4.1 ± 0.2	4.6 ± 0.1
C11:0	0.10 ± 0.00	0.2 ± 0.1	0.2 ± 0.1	0.2 ± 0.1
C12:0	5.5 ± 0.1 ^b^	5.3 ± 0.1 ^b^	5.2 ± 0.2 ^b^	5.6 ± 0.1 ^a^
C13:0	0.12 ± 0.01	0.10 ± 0.00	0.10 ± 0.01	0.11 ± 0.02
C14:0	14.62 ± 0.02	14.3 ± 0.2	14.2 ± 0.3	14.1 ± 0.9
C14:1	1.15 ± 0.02	1.18 ± 0.06	1.20 ± 0.07	1.11 ± 0.04
C15:0	1.18 ± 0.00 ^b^	1.17 ± 0.02 ^b^	1.14 ± 0.03 ^b^	1.23 ± 0.01 ^a^
C15:1	0.25 ± 0.01 ^b^	0.27 ± 0.01 ^a,b^	0.26 ± 0.01 ^b^	0.28 ± 0.01 ^a^
C16:0	39.34 ± 0.02 ^b^	38.4 ± 0.3 ^c^	38.9 ± 0.5 ^c^	39.9 ± 0.2 ^a^
C16:1	1.57 ± 0.01 ^a^	1.61 ± 0.06 ^a^	1.57 ± 0.08 ^a^	1.38 ± 0.07 ^b^
C17:0	0.61 ± 0.02 ^b^	0.61 ± 0.02 ^b^	0.61 ± 0.05 ^b^	0.70 ± 0.05 ^a^
C18:0	5.71 ± 0.02	5.64 ± 0.06	5.7 ± 0.1	5.74 ± 0.09
C18:1n-9	18.4 ± 0.1 ^b^	19.4 ± 0.3 ^a^	19.8 ± 0.3 ^a^	18.0 ± 0.7 ^a^
C18:2n-6	0.79 ± 0.03	1.4 ± 0.3	1.52 ± 0.07	0.7 ± 0.7
C18:3n-3	0.023 ± 0.001 ^b^	0.123 ± 0.003 ^a^	0.12 ± 0.00 ^a^	0.15 ± 0.01 ^a^
C20:0	0.21 ± 0.01	0.32 ± 0.03	0.31 ± 0.06	0.32 ± 0.08
SFA	77.81 ± 0.07 ^b^	76.1 ± 0.5 ^a^	75.6 ± 0.3 ^a^	78.5 ± 1.2 ^a^
MUFA	21.37 ± 0.09 ^b^	22.4 ± 0.3 ^a^	22.8 ± 0.3 ^a^	20.7 ± 0.7 ^b^
PUFA	0.82 ± 0.03	1.4 ± 0.4	1.6 ± 0.1	0.8 ± 0.6

SFA—saturated fatty acids; MUFA—monounsaturated fatty acids; PUFA—polyunsaturated fatty acids. ^a–c^ Means within the same row with different superscript letters are significantly different (*p* < 0.05).

## Data Availability

Data available on request due to restrictions e.g., privacy or ethical. The data presented in this study are available on request from the corresponding author.

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
