# Peer review of "Characterization of Kefir Produced in Household Conditions: Physicochemical and Nutritional Profile, and Storage Stability"

_foods, 2021, doi:10.3390/foods10051057_

Round 1

Reviewer 1 Report

Report on the article Foods-1197185

The article studies the physicochemical and nutritional characteristics of kefir produced by fermentation of semi-skimmed UHT-treated milk using CIDCA AGK1 grains.

MAJOR POINTS

The work does not present absolutely any innovation, neither for the raw materials used, nor for the methodology used nor for the results obtained, and it only confirms results repeatedly obtained and previously discussed in other studies reported in the literature.

The authors indicate as a particularity of the product, highlighting it even in the title, the fact that it is obtained from Portuguese milk. What differentiating characteristics does Portuguese cow's milk have in relation to cow's milk from other countries?

Obviously, apart from the microbial composition of kefir grains, grain to milk ratio, fermentation conditions, etc., the composition and nutritional quality of kefir depends on the composition of the milk used as raw material. In this study, composition of milk was not studied or reported. Neither has any physicochemical properties of the milk used been studied. It is only indicated that it is semi-skimmed UHT-treated milk.

The authors indicated in the abstract of the article that “This work opens the possibility of using the reported protocoI to prepare and store kefir…”. What is the novelty of this protocol in relation to the preparation and storage protocols previously used?

OTHER POINTS

-Line 184: “Man, Rogosa, and Sharp” should be “de Man, Rogosa and Sharpe”.

Author Response

Manuscript ID: foods entitledPhysicochemical and nutritional characterization of kefir obtained from Portuguese milk fermented with CIDCA AGK1 grains and stability studies during storage”

Dear Reviewer 1:

We appreciate the comments of reviewer 1 and we follow them as main contributions to the improvement of the manuscript. We carefully considered all suggestions, which we addressed and incorporated into the manuscript as detailed bellow. The manuscript has been reviewed for English language. Changes to the corrected paper have a blue background.

Reviewer 1: Detailed comments: 

MAJOR POINTS

Comment 1: The work does not present absolutely any innovation, neither for the raw materials used, nor for the methodology used nor for the results obtained, and it only confirms results repeatedly obtained and previously discussed in other studies reported in the literature.

And

Comment 4: The authors indicated in the abstract of the article that “This work opens the possibility of using the reported protocoI to prepare and store kefir…”. What is the novelty of this protocol in relation to the preparation and storage protocols previously used?

Author’s answer: We thank the reviewer for the comments. We accept all the comments, so we changed the manuscript and incorporated it into the text as suggested.

The innovation of our work is based on the simulation of household conditions, due to the fact that these are the conditions our team wants to replicate in the continuity of the study.

Line 90: “This work aimed to study kefir in representative household conditions, characterizing the properties, nutritional composition and stability of a freshly made and 48 h refrigerated beverage. To date, information on homogeneity and stability of traditional kefir is scarce, and to the best of our knowledge, this is the first study to provide information on kefir produced by simulating representative home use conditions. The innovative character of this study also resides in the fact that, in addition to the usual parameters for physicochemical analysis and composition, the fatty acid profile, particle size, PdI, zeta potential and FTIR spectra analysis were also included in the global evaluation of kefir."

Comment 2: The authors indicate as a particularity of the product, highlighting it even in the title, the fact that it is obtained from Portuguese milk. What differentiating characteristics does Portuguese cow's milk have in relation to cow's milk from other countries?

Author’s answer: We thank the reviewer for the comments. In reality, we aimed to study the effect of representative household conditions, in the production of kefir and in its 2-days storage. So, we changed the manuscript and incorporated it into the text as suggested.

Line 2 - Title: “Characterization of kefir produced in household conditions: physicochemical and nutritional profile and storage stability”

Comment 3: Obviously, apart from the microbial composition of kefir grains, grain to milk ratio, fermentation conditions, etc., the composition and nutritional quality of kefir depends on the composition of the milk used as raw material. In this study, composition of milk was not studied or reported. Neither has any physicochemical properties of the milk used been studied. It is only indicated that it is semi-skimmed UHT-treated milk.

Author’s answer: We thank the reviewer for the comments. We would like to draw attention to tables 2, 3, 4 and 5 and to figure 3, were the respective composition of the milk (control) is showed.

Line 252: “Table 2 - Hydrodynamic diameter, PdI, and zeta potential of control and kefir samples”

Line 264: “Figure 3 - FTIR spectra of control and kefir samples”

Line 278: “Table 3 - Physical parameters of control and kefir samples”

Line 295: “Table 4 – Nutritional composition of control and kefir samples”

Line 320: “Table 5 – Fatty acids profile of control and kefir samples”

OTHER POINTS

Comment 1: Line 184: “Man, Rogosa, and Sharp” should be “de Man, Rogosa and Sharpe”.

Author’s answer: We thank the reviewer for the correction. We changed the manuscript and incorporated it into the text as suggested.

now Line 142: “LAB counts were quantified on De Man, Rogosa and Sharpe”

Author´s comment: Considering the reviewer suggestions, we changed the manuscript as follows:

Introduction, discussion and conclusion sections were reviewed in order to provide enough background and to be sufficiently supported by the results, respectively.

Reviewer 2 Report

Dear authors, your paper on "Physicochemical and nutritional characterization of kefir obtained from Portuguese milk fermented with CIDCA AGK1 grains and stability studies during storage" is in my opinion of limited originality. CIDCA AGK1 grains have been used and characterized in many other studies. Although it is the first time that they are used to produce an household kefir starting from UHT milk, the differences, if any, should have been presented as compared to different type of milk. The characterization of the kefir to bu further used in the project DermapBio might be necessary, but does not necessarily worth a separate pubblication. Some preliminary indication on the functionality of the produced kefir related to the analysed characteristics should have be presented.

Following some specific comments: 

the introduction section is too long in my opinion. It looks more like a review paper than an original research.

line 53: a new classification of lactic acid bacteria has been accepted by the literature after Zheng et al. 2020 paper. Saccharomyces has a typo.

Line 94: which yeasts produce acetic acid? 

line 122: the lipolytic activity of LAB is known to be limited 

line 140: should be re-phrased. Also, I think that meet the codex requirements whould be important for a commercial production. The product that you are making is ment to be used in a future research. 

line 164: rinsing

line 168: The weighted kefir was then employed for a new inoculum? If yes, the filter was sterile and the sterility maintained?

line 295: lacti acid

line 333: the naming of the samples is a bit confusing: T= refers to 24 hours of fermentation, T24 to 24 of storage..

line 350: lactic acid 

line 361: this is not really in contrast, it is expected that when lactose is consumed, lacti acid is produced..

line 404: no information is given on the potential functionality of the produced and analyzed kefir on the cutaneous effects

line 432: this cannot be said actually, as no microbial characterization of the grains during time is monitored

line 445: the particle of the grains is not only linked to casein aggregation...

line 485: the viscosity can be linked to the EPS but this is probably not the unique reason. EPS have not been quantified. 

line 493: viscosity and syneresys usually occur during storage..you measured casein aggregation (synerisis) but did not observe viscosity reduction..how do you comment on this?

line 501: 11 % g/100g ??

line 504: this is not clear to me..

line 514: throughout

line 516: lactic acid

ine 540: Proteins form aggregates, are denatured and undergoes to many transformations, so maybe is not possible to find differences in the total nitrogen present, but this doesn't mean that proteins remains the same. Moreover how can you say that caseins doesn't change if you don't have analyzed them?

line 567: to say this you should have a correlation between microbial profile and FA. 

line 585: as I said, this is not fully justified by the presented results in my opinion

Author Response

Manuscript ID: foods entitledPhysicochemical and nutritional characterization of kefir obtained from Portuguese milk fermented with CIDCA AGK1 grains and stability studies during storage”

Dear Reviewer 2:

We appreciate the comments of reviewer 2 and we follow them as main contributions to the improvement of the manuscript. We carefully considered all suggestions, which we addressed and incorporated into the manuscript as detailed bellow. The manuscript has been reviewed for English language. Changes to the corrected paper have a yellow background.

Reviewer #2: Detailed comments: 

Comment 1: the introduction section is too long in my opinion. It looks more like a review paper than an original research.

Author’s answer: We thank the reviewer for the comment. Introduction section was reviewed in order to become shorter as suggested.

Comment 2 Line 53: a new classification of lactic acid bacteria has been accepted by the literature after Zheng et al. 2020 paper. Saccharomyces has a typo.

Author’s answer: We thank the reviewer for the correction. Suggested literature was consulted and incorporated in the text. Corrections were incorporated into the text as suggested.

Now Line 45: “Lactobacillaceae family (genera Lactobacillus and Leuconostoc) and Streptococcaceae family (genera Lactococcus and Streptococcus), acetic acid bacteria Acetobacteraceae family (genera Acetobacter) and yeasts Saccharomycetaceae family (genera Kluyveromyces and Saccharomyces)”

Comment 3 Line 94: which yeasts produce acetic acid? 

Author’s answer: We thank the reviewer for the correction. Acetic acid is a physiological product of Saccharomyces cerevisiae fermentation (Front. Microbiol., 20 February 2013 | https://doi.org/10.3389/fmicb.2013.00033). In kefir, acetic acid may be produced either by LAB, by acetic acid bacteria or by yeasts (S. cerevisiae), however the main and product of yeast fermentation is ethanol.  We changed the manuscript and incorporated it into the text as suggested.

Now Line 76: “The production of lactic acid contributes to the antimicrobial effect of kefir, and since it acts as natural preservative, allows the homemade product to have a low contamination risk”

Comment 4 line 122: the lipolytic activity of LAB is known to be limited 

Author’s answer: We thank the reviewer for the correction. We changed the manuscript and incorporated it into the text as suggested.

Now Line 88: “Although the lipolytic activity in milk fat by LAB is limited, it can still contribute to the production of free fatty acids”

Comment 5 line 140: should be re-phrased. Also, I think that meet the codex requirements whould be important for a commercial production. The product that you are making is meant to be used in a future research. 

Author’s answer: We thank the reviewer for the comment. Although we meant to have a traditional kefir production, not a commercial one, we considered that compliance with codex requirements reinforced the safe usability of the product. Considering the reviewer comment we withdraw this phrase from the introduction.

Comment 6 line 164: rinsing

Author’s answer: We thank the reviewer for the correction. Correction was made in line 164.

Now Line 124: “Grains were rinsed with milk at room temperature”

Comment 7 line 168: The weighted kefir was then employed for a new inoculum? If yes, the filter was sterile and the sterility maintained?

Author’s answer: We thank the reviewer for the correction. We changed the manuscript and incorporated it into the text as suggested. Because it was our intention to replicate household conditions to produce homemade kefir, we did not use sterile equipment, nevertheless all food security requirements were assured.

Now Line 127: “After weighing, kefir grains were used as a new inoculum, maintaining the grain to milk ratio.”

Comment 8 lines 295, 350 and 516: lactic acid

Author’s answer: We thank the reviewer for the correction. We changed the manuscript and incorporated it into the text as suggested.

Now Lines 238, 282 and 451: “lactic acid”

Comment 9 line 333: the naming of the samples is a bit confusing: T= refers to 24 hours of fermentation, T24 to 24 of storage..

Author’s answer: We thank the reviewer for the correction. Kefir samples nomenclature were reviewed and incorporated into the text as suggested.

Marked throughout the text: kefir immediately after fermentation = “t0” and kefir after storage at 5 ± 1 °C for 24 and 48 h (t24 and t48, respectively).

Comment 10 line 361: this is not really in contrast, it is expected that when lactose is consumed, lactic acid is produced..

Author’s answer: We thank the reviewer for the correction. We changed the manuscript and incorporated it into the text as suggested.

Now Line 307: “Consequently the lactic acid content in fresh kefir”

Comment 11 line 404: no information is given on the potential functionality of the produced and analyzed kefir on the cutaneous effects

Author’s answer: We thank the reviewer for the correction. Because effects of traditional kefir intake on skin is the aim of our team project, we decided not to incorporate this item in this manuscript. We change the manuscript as follows.

Now Line 340: “The traditional use, combined with the fact that 20 °C is in the range of typical indoor conditions of a Portuguese house [52], thus reflecting the domestic scenario preparation of kefir, justifies the choice of the fermentation temperature in our study.”

Comment 12 line 432: this cannot be said actually, as no microbial characterization of the grains during time is monitored

Author’s answer: We thank the reviewer for the correction. We changed the manuscript and incorporated it into the text as suggested.

Now Line 373: “and may reflect the LAB capability to acidify the milk”

Comment 13 line 445: the particle of the grains is not only linked to casein aggregation...

Author’s answer: We thank the reviewer for the comment. We refer that it was the particle size of kefir, not kefir grains, that are related, but not exclusively, to the casein aggregation, so we changed the manuscript and incorporated it into the text as suggested.

Now Line 387: “thus these factors may directly affect particle growth in kefir beverage”

Comment 14 line 485: the viscosity can be linked to the EPS but this is probably not the unique reason. EPS have not been quantified. 

Author’s answer: We thank the reviewer for the correction. We changed the manuscript and incorporated it into the text as suggested.

Now Line 435: “This can be in part attributed to the production”

Comment 15 line 493: viscosity and syneresys usually occur during storage..you measured casein aggregation (synerisis) but did not observe viscosity reduction..how do you comment on this?

Author’s answer: We thank the reviewer for the comment. We changed the manuscript and incorporated it into the text as suggested.

Now Line 443: “however, these changes only become evident in periods of storage longer than seven days [34,36,44]. Nevertheless, our data showed no difference in viscosity during storage, which may be attributed to the limited storage time (only 48 hours).”

Comment 16 line 501: 11 % g/100g ??

Author’s answer: We thank the reviewer for the correction. We changed the manuscript and incorporated it into the text as suggested.

Now Line 437: “11.9 g/100g”

Comment 17 line 504: this is not clear to me..

Author’s answer: We thank the reviewer for the comment. We changed the manuscript and incorporated it into the text as suggested.

Now Line 451: “Whilst typical cow milk presents a carbohydrate content between 4.7 and 4.9 g/100 mL, reflecting essentially lactose content [68], kefir as carbohydrate content around 11.9 g/100g, also reflecting the presence of polysaccharide kefiran”

Comment 18 line 514: throughout

Author’s answer: We thank the reviewer for the correction. We changed the manuscript and incorporated it into the text as suggested.

Now Line 465: “throughout”

Comment 20 line 540: Proteins form aggregates, are denatured and undergoes to many transformations, so maybe is not possible to find differences in the total nitrogen present, but this doesn't mean that proteins remains the same. Moreover how can you say that caseins doesn't change if you don't have analyzed them?

Author’s answer: We thank the reviewer for the comment. We changed the manuscript and incorporated it into the text as suggested.

Now Line 485: “producing different peptides and nonprotein nitrogen compounds, thus contributing to the protein profile of kefir”

Now Line 489: “Even though the protein profile has not been determined in our work, its results are hereby supported, since no differences in the total protein content of kefir and unfermented milk were observed (Table 4).”

Comment 21 line 567: to say this you should have a correlation between microbial profile and FA. 

Author’s answer: We thank the reviewer for the comment. We accept the suggestion, but we did not conclude about the relation between microbial profile and FA based on our results, on the contrary we presented results different from ours and the respective justifications made by those authors. We changed the manuscript and incorporated it into the text as suggested.

Now Line 515: “The differences observed in kefir´s fatty acids profiles, according to other authors, may be justified by the different origin of the grains since each bacterial community may present unique fatty acids production”

Comment 22 line 585: as I said, this is not fully justified by the presented results in my opinion

Author’s answer: We thank the reviewer for the comment. We changed the manuscript and incorporated it into the conclusions as suggested.

Now Line 519: “Our results showed that the kefir produced under home use conditions using UHT milk is able to fulfill the Codex Alimentarius requirements and maintains its characteristics with respect to the physicochemical composition, both after fermentation, as well as during 48 hours of refrigerated storage. Whereas fat, protein and carbohydrate content suffered no significant changes over fermentation, lactic acid increased, and lactose decreased, as expected. The fatty acids profile of the milk and kefir samples changed during fermentation revealing a decrease in SFA, an increase in MUFA, and no change in PUFA. Refrigerated storage did not significantly impact nutritional composition and fatty acids profile, thus attesting for the stability of kefir under these conditions.

To the best of our knowledge, this is the first study to aggregate information on detailed composition, homogeneity and stability after refrigeration, of kefir produced using CIDCA AGK1 grains in a traditional in use setting. This work further contributes to the characterization of this food that is so widely consumed around the world by focusing on kefir that was produced in typical home use conditions.”

Round 2

Reviewer 1 Report

This reviewer reaffirms the opinion that the article barely increases the scientific knowledge in this topic. However, the manuscript has been substantially improved in relation to the initial submission.

Reviewer 2 Report

I thank the authors for having considered my suggestions.